# Preparation and Characterization of Au/NiPc/Anti-p53/BSA Electrode for Application as a p53 Antigen Sensor

Yen-Jou Chen [1], Yu-Ren Peng [1], Hung-Yu Lin [1], Tsung-Yu Hsueh [1], Chao-Sung Lai [2,3,*] and Mu-Yi Hua [1,4,*]

1 Department of Chemical and Materials Engineering, Biomedical Engineering Research Center, and Healthy Aging Research Center, Chang Gung University, Taoyuan 333323, Taiwan; zoejouyie@gmail.com (Y.-J.C.); elipeng@gapp.nthu.edu.tw (Y.-R.P.); zcbm8217@gmail.com (H.-Y.L.); skyboy60314@gmail.com (T.-Y.H.)
2 Department of Electronic Engineering, Biomedical Engineering Research Center, and Artificial Intelligent Research Center, Chang Gung University, Taoyuan 333323, Taiwan
3 Department of Nephrology, Chang Gung Memorial Hospital, Taoyuan 333005, Taiwan
4 Department of Gastroenterology and Hepatology, Linkou Chang Gung Memorial Hospital, Taoyuan 333423, Taiwan
* Correspondence: cslai@mail.cgu.edu.tw (C.-S.L.); huamy@mail.cgu.edu.tw (M.-Y.H.); Tel.: +886-3-2118800 (ext. 5750) (C.-S.L.); +886-3-2118800 (ext. 5289) (M.-Y.H.)

**Abstract:** While the tumor suppressor protein p53 regulates the cell cycle to prevent cell damage, it also triggers apoptosis and prevents cancer. These inhibitory functions may disappear once the p53 gene is mutated. Under these circumstances, the detection of p53 protein concentrations can have significant clinical applications. In this study, nickel phthalocyanine (NiPc) was coated on a gold electrode to produce a modified Au/NiPc electrode. p53 antibodies were bonded to the Au/NiPc electrode by the $Ni^{+2}$ ion in NiPc, which can be self-assembled with the imidazole group of the p53 protein. The Au/NiPc/anti-p53 electrode was subsequently dripped with a buffer solution of bovine serum albumin (BSA) to form the Au/NiPc/anti-p53/BSA electrode, which was used for the detection of p53 antigen under 10 mM potassium ferricyanide/potassium ferrocyanide ($K_3Fe(CN)_6/K_4Fe(CN)_6$) solution by cyclic voltammetry and differential pulse voltammetry analyses. The linear detection range and the sensitivity for the p53 antigen were 0.1–500 pg/mL and 60.65 μA/Log (pg/mL)-$cm^2$, respectively, with a detection time of 90–150 s. In addition, Au/NiPc/anti-p53 (100 ng/mL)/BSA electrodes were tested for specificity using glucose, bovine serum albumin, histidine, ascorbic acid, uric acid, prostate-specific antigen, human serum albumin, and human immunoglobulin G. All *p*-values were <0.0005, indicating an outstanding specificity.

**Keywords:** tumor suppressor protein; nickel phthalocyanine; p53 protein; differential pulse voltammetry; cyclic voltammetry; cancer biomarker

## 1. Introduction

According to the World Health Organization (WHO), cancer was the second leading cause of death in the world in 2018 [1], and early diagnosis and treatment of malignancies can significantly increase survival rates. Tumor biomarkers can provide effective and timely information in early cancer monitoring [2]; hence, early diagnosis is one of the key factors to ensure successful clinical outcomes.

The tumor suppressor protein p53 is the protein associated with the viral cancer gene detected and classified in the field of cancer biology by Lane and Crawford in a study of converted cells and tumors in 1979 [3]. The p53 gene family and its proteins are essential in preventing the malignant transformation of normal cells into cancer cells. It is known as the guardian of the genome, located on chromosome 17 in humans. Its coding gene is Tp53, and p53 is a transcription factor that is responsible for controlling cell division, DNA repair, and cancer suppression genes, ultimately playing a vital role in preventing gene mutations. During transformation of human cells, p53 is initially activated by a series

of cell signals, including undernourishment, hypoxia, and activation of cancer-causing genes, thus preventing DNA damage during the G1/S phase of the cell cycle G1/S [4]. Failure to repair DNA damage can trigger apoptosis, which prevents cells with abnormal genetic information from continuing to divide and growing into tumors [5]. Once the p53 gene is mutated, these inhibitory functions are impaired. Therefore, p53 can be considered as a switch for the mechanism of cancer suppression. Clinically, the p53 protein in the normal human body has been shown in most studies to have a half-life of only about 5 to 20 min. In presence of mutations, protein degradation, or combinations with viral cancer genes, the half-life of the p53 protein is increased; in turn, this event induces the immune system to produce p53 autoantibodies [6]. As a result, there is a clear correlation between the concentration of p53 antibodies in the body and tumor size, cell differentiation level, and lymph node metastasis [2]. The p53 antibodies can be used as a prognostic and tracking indicator for lung, breast, head, and neck cancers. They also show a high diagnostic performance for various solid malignancies—including rectal cancer [7], prostate adenocarcinoma [8], cervical cancer, and oral cancer [9].

In recent years, a variety of techniques have been developed to detect protein biomarkers, including enzyme-linked immunosorbent assays (ELISA) [10], DNA probe technology [11], immunofluorescent assays [12], and electrochemical luminescent immunoassays [13]. Among them, electrochemical analysis has been widely used because of its ease-of-use, time-saving capability, high sensitivity, and high selectivity [14]. In 2004, Yan et al. used immunomagnetic electrochemical luminescence analysis to quantify p53 concentrations in serum samples obtained from cancer patients [15]. In 2016, Pedrero et al. used mesh-printed carbon electrodes to secure carbide-based beads with magnets; they subsequently captured the p53 protein with the 1-ethyl-3-(3-dimethylaminopropyl) carbodiimide/N-hydroxysuccinimide (EDC/NHS) joint and rated the concentration of p53 antibodies with the spicy root hydrogen peroxide enzyme [16]. In 2017, Giannetto et al. studied untreated and diluted urine samples for urinary malignancies, modified mesh carbon electrodes with nanotubes/nanoparticles, and fixed p53 antibodies to the modified electrodes to sense the p53 protein [17]. In 2020, Kang et al. conducted a study with high molecular poly(3,4-ethylenedioxythiophene):polystyrenesulfonate (PEDOT:PSS)/ nano-gold grains modified on glass carbon electrodes and devised a zeolitic imidazolate framework-8/2 (ZIF-8/2), 3-diaminophenazine (DAP)/two-resistance sandwich electrochemical immunoassay for detecting and quantifying p53 proteins [14].

Nickel phthalocyanine (NiPc) has recently been used as an electron transfer medium [18] for thin-film applications in solar cells [19], gases [20], and electrochemical sensors because of its outstanding electrochemical properties and chemical stability [21]. In this study, p53 antibodies were bonded to a NiPc-modified gold electrode by the $Ni^{+2}$ ion of NiPc, which can be self-assembled with the imidazole group of the protein, and the p53 antigen was detected at different concentrations using the modified electrodes. The current change relationship was rapidly detected by electrochemical cyclic voltammetry (CV) and differential pulse voltammetry (DPV) to obtain the concentration detection curve of p53, which can be used as a sensor for quantifying p53 antigen concentrations in the serum.

## 2. Materials and Methods

### 2.1. Apparatus and Reagents

An electrochemical analysis was carried out using an electrochemical measurement system (DropSens sStat 8000). In total, three different electrode-type commercial screen-printed gold electrodes (G3-Flat Gold electrode) were used in this study; their working electrode area was 23.75 $mm^2$. The working, corresponding, and reference electrodes were NiPc-modified gold, carbon, and Ag/AgCl electrodes, respectively. Experimental conditions for CV were 20 cycles at a scanning rate of 50 mV/s from −0.2 V to 0.5 V. The differential pulse voltammetry (DPV) was measured at an amplitude of 0.05 V between 0 V and 0.4 V. Qualitative analysis was performed by detecting the liquid and solid absorption spectra of the material with ultraviolet/visible near-infrared absorption spectrometer at

330–900 nm (JASCO V/u2012650). The functional group analyses of the materials were performed using a Fourier transfer infrared spectrometer (Bruker Tensor 27 FTIR spectrometer) at 4000–500 cm$^{-1}$. NiPc was obtained from Acros Organics (Code: 415600250), tetrahydrofuran (THF), ascorbic acid (AA), uric acid (UA), potassium ferricyanide ($K_3Fe(CN)_6$), potassium ferrocyanide ($K_4Fe(CN)_6$) from Merck, glucose, histidine from Thermo, tumor suppressor gene antibody (anti-p53 antibody, ab1101), tumor suppressor gene recombinant protein (human p53 protein, ab43615), bovine serum albumin (BSA) from Elabscience (E-IR-R108), prostatic specific antigen (PSA) from Abcam, human serum albumin (HSA) from Cusabio, and immunoglobulin G (IgG) from Sigma. Phosphate-buffered saline (PBS) was used as a dilution solution, configured by adjusting the HCl or NaOH concentration of 1 M $KH_2PO_4$, 136.9 mM NaCl, 2.7 mM KCl, and 8.1 mM $Na_2HPO_4$ to pH = 7.4 with deionized (DI) water.

### 2.2. Film Fabrication Process

### 2.2.1. Purification of NiPc

NiPc (purity > 95%) was dissolved in chloroform into 1 M HCl aqueous solution several times using a fractional funnel and the water layer was subsequently filtered. The extraction procedure was repeated 10 times with DI water. After extraction, NiPc was dissolved in chloroform and vacuum-dried. The improvement in the purity of NiPc greatly increased the stability of the NiPc-modified gold electrode and its electrochemistry.

### 2.2.2. Preparation of Au/NiPc Electrode

The commercial screen-printed gold electrodes (G3-Flat Gold electrode) were vacuum-packed before use. Before modification with NiPC, the Au working electrode was cleaned three times (1 min for each time) with THF (20 mL). After three sequential washes with deionized water (20 mL) and acetone (20 mL), the Au electrode was vacuum-dried for 1 h at 25 °C. NiPc was dissolved in a THF solution of $2.5 \times 10^{-3}$–$25 \times 10^{-3}$ wt%. NiPc/THF solution (2 μL) was cast to the Au working electrode and dried in a vacuum for 1 h at 25 °C. Subsequently, 200 μL droplets of 10 mM $K_3Fe(CN)_6$)/($K_4Fe(CN)_6$ solution were added to the electrode surface for CV and DPV analyses.

### 2.3. Construction of the Electrochemical Sensor

The experimental study design is summarized in Figure 1. Incubations were carried out at 4 °C to reduce the risk of denaturation for both p53 antigens and anti-p53 antibodies. First, 20 μL anti-p53 (1 ng/mL and 100 ng/mL, respectively) was dropped on the dry Au/NiPc electrode surface and incubated for 1 h at a temperature of 4 °C. The electrode was then washed with DI water to obtain an Au/NiPc/anti-p53 (1 ng/mL) and Au/NiPc/anti-p53 (100 ng/mL) electrode. Both of the Au/NiPc/anti-p53 electrodes surface were then dripped with 20 μL BSA (136 μg/mL) solution as a protein blocker and incubated for 1 h at a temperature of 4 °C. Then, the non-bonded BSA was washed away with DI water to obtain the Au/NiPc/anti-p53 (1 ng/mL)/BSA and Au/NiPc/anti-p53 (100 ng/mL)/BSA electrodes. For each fixation step, 200 μL drops of 10 mM $K_3Fe(CN)_6$)/($K_4Fe(CN)_6$ solution were added to the Au/NiPc/anti-p53 (1 ng/mL)/BSA and Au/NiPc/anti-p53 (100 ng/mL)/BSA electrodes surface for CV and DPV analyses.

### 2.4. Detection of p53 Concentrations

The p53 protein solutions were prepared at different concentrations (0.1–500 pg/mL) in PBS. Subsequently, 20 μL drops of each solution were added to the Au/NiPc/anti-p53 (1 ng/mL)/BSA and Au/NiPc/anti-p53 (100 ng/mL)/BSA electrodes and incubated for 1 h at 4 °C. The unticked p53 protein was washed off with DI water. Finally, 200 μL of 10 mM $K_3Fe(CN)_6$)/($K_4Fe(CN)_6$ solution were added to the electrode surface for CV and DPV analyses to establish a calibration curve for p53 concentrations.

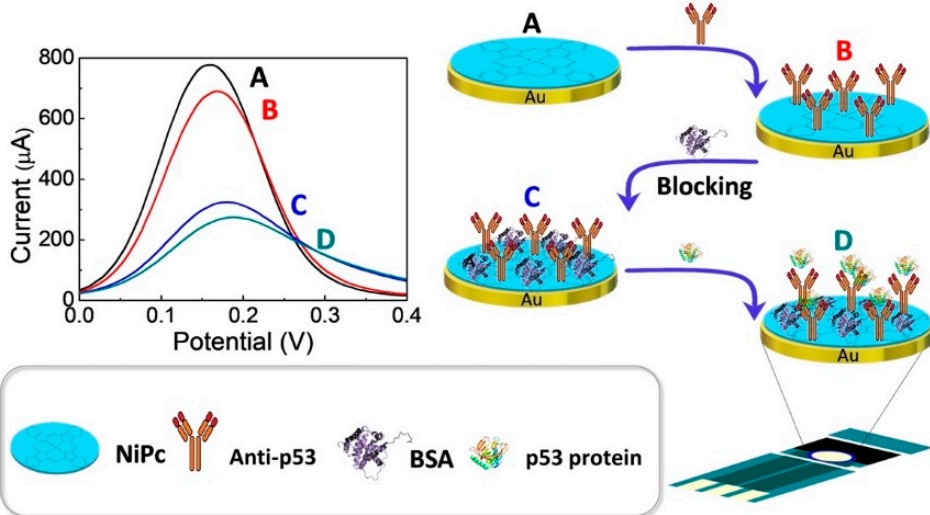

**Figure 1.** Experimental design of the electrochemical sensor. DPV curves of (**A**) Au/NiPc electrode, (**B**) Au/NiPc/anti-p53 electrode, (**C**) Au/NiPc/anti-p53/BSA electrode, and (**D**) Au/NiPc/anti-p53/BSA/VEGF electrodes.

### 2.5. Specificity Test of p53 by Au/NiPc/anti-p53(100 ng/mL)/BSA Tlectrode

This experiment was conducted with the following serum-specific compounds: 0.6 mg/dL ascorbic acid (AA), 5 mg/dL uric acid (UA), 100 mg/dL glucose, 0.136 mg/mL BSA, 1.9 mg/dL histidine, 4 ng/mL prostate-specific antigen (PSA), 6.57 mg/mL HSA, and 1.0 mg/mL human immunoglobulin G (IgG). The goal was to mimic the composition of human serum in an effort to analyze the specificity of electrodes. The interference drops (20 μL) were added to the Au/NiPc/anti-p53/BSA electrode and placed in an environment with a temperature of 4 °C. After 1 h, the non-bonded analytes were washed away with DI water, and then 200 μL of 10 mM $K_3Fe(CN)_6$/$(K_4Fe(CN)_6$ solution to the electrode surface for CV and DPV analyses. The specificity analysis was conducted as a statistical comparison of differences between the interfering compounds and the p53 antigens detected with Au/NiPc/anti-p53 (100 ng/mL)/BSA electrodes. A *p*-value was a measure of the probability that an observed difference could have occurred just by random chance. When $p < 0.05$, the difference between the two samples was statistically significant; conversely, there was no statistical difference when *p* was $> 0.05$ [22].

## 3. Results and Discussion

### 3.1. Material Characterization

#### 3.1.1. Ultraviolet-Visible and Near-Infrared (UV-Vis Near IR) Analysis

The UV-Vis near IR spectra of the NiPc/CHCl$_3$ solution and the solid film of NiPc are shown in Figure 2. From Figure 2a, it can be seen that there are three main absorption peaks at 350, 634, and 664 nm. While peaks in the visible region of 600–700 nm are in the Q band, the peak at 350 nm is in the B-band of 300–500 nm, which correspond to the π–π* transition [23]. The B-band, also known as the Soret band, arises mainly due to the exchanging of internal and external electrons, while the Q band arises due to the external electrons of the benzene rings on the porphyrins ring [24]. From Figure 2b, the red-shifts of all absorption peaks to higher wavelengths of the solid-state membrane can be seen, which is due to the tighter stacking of NiPc molecules in the solid film compared to the liquid. This in turn leads to an increase in the common plane and a decrease in the degree of freedom between molecules.

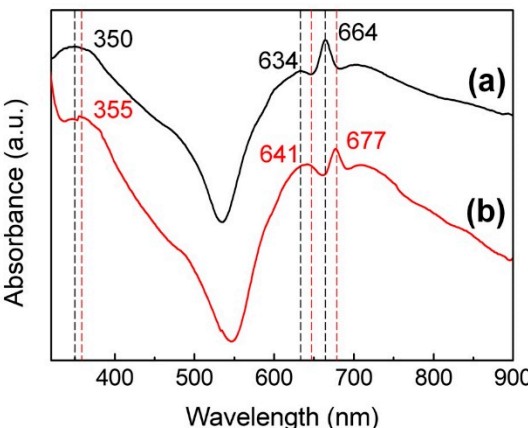

**Figure 2.** UV-Vis near IR spectra of the (**a**) NiPc/CHCl$_3$ solution and the (**b**) solid film of NiPc.

### 3.1.2. Fourier-Transform Infrared Spectroscopy Analysis

In order to verify that the protein self-assembled with NiPc, the NiPc/BSA powder was obtained by mixing BSA and NiPc in DI water for 3 h before drying. The FTIR spectra of BSA, NiPc, and NiPc/BSA are shown in Figure 3, with the significance of individual characteristic peaks listed in Table 1. As shown in Figure 3a, the main characteristic peaks of BSA are 1539 cm$^{-1}$ (due to N-H bending and C-N stretch vibration), 1658 cm$^{-1}$ (due to C=O stretch vibration), and 3296 cm$^{-1}$ (due to O-H and N-H stretching vibration) [25,26]. From Figure 3b, the main characteristic peaks of NiPc are due to metal lids, pyrroles, and benzene rings, where the characteristic peak for 916 cm$^{-1}$ is due to the vibration of Ni$^{+2}$ with lids, 1290 cm$^{-1}$ and 1429 cm$^{-1}$ are C-N and C-C stretching vibrations from the pyrrole rings, respectively, and 1122 cm$^{-1}$ corresponds to bending vibration (in-plane bending) in the benzene ring [21,23,25–28]. As shown in Figure 3c, in addition to the NiPc spectral peaks, NiPc/BSA generates three new peaks at 3296 cm$^{-1}$, 1653 cm$^{-1}$, and 1533 cm$^{-1}$, which are contributed by BSA. Comparing Figure 3a,c, it can be seen that the N-H bending and C-N stretching of BSA shifted from 1539 cm$^{-1}$ to 1533 cm$^{-1}$ and became stronger and wider. This could result from Ni$^{+2}$ ion of NiPc producing self-assembled bonding, allowing N-H bending vibration and C-N stretching vibration from BSA to shift to a lower frequency [27,29]. These three newly generated characteristic peaks demonstrate that BSA protein molecules were bound to NiPc.

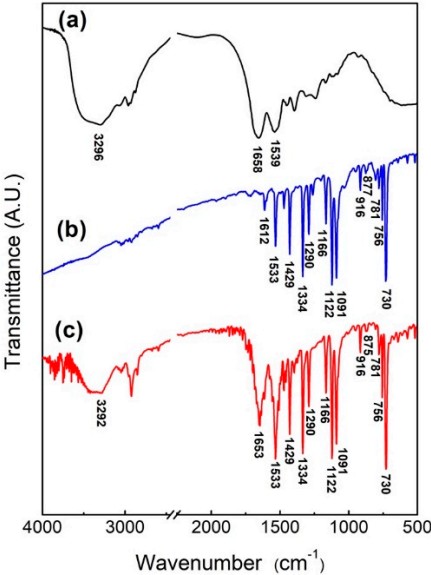

**Figure 3.** FTIR spectra of (**a**) BSA, (**b**) NiPc, and (**c**) NiPc/BSA.

**Table 1.** Characterized peaks of FTIR of NiPc and BSA [21,23,25–28].

| Wavelength (cm$^{-1}$) | Assignment |
|---|---|
| **BSA** | |
| 1539 | Coupling of bending vibration of N-H and stretching vibration of C–N [25] |
| 1658 | C=O stretching vibration [25] |
| 3296 | N-H/O-H stretching vibration [26] |
| **NiPc** | |
| 730 | C-H out-of-plane angular deformation [21] |
| 756 | Pc ring, C-N=C pyrrole stretching vibration [21] |
| 781 | C-H out-of-plane angular deformation benzene breathing [21] C-N out-of-plane angular deformation [25] |
| 877 | C-H out-of-plane angular deformation [21] |
| 916 | Metal (Ni) ligand vibration [27] |
| 1091 | in-plane C-H deformation [28] |
| 1122 | in-plane bending vibration in benzene ring [23] |
| 1166 | C-N bending vibration [28] |
| 1290 | C-N stretching vibration in pyrroles [23] |
| 1334 | pyrrole stretching vibration [21] |
| 1429 | C-C bonds stretching vibration in pyrroles [23] |
| 1533 | isoindol stretching vibration [21] |
| 1612 | benzene ring stretching vibration [27] |

*3.2. Optimal Conditions for Electrode Preparation*

3.2.1. NiPc Concentration for the Film Coating

The DPV analyses for the preparation of Au/NiPc electrodes at different NiPc/THF concentrations ($2.5 \times 10^{-3}$, $5 \times 10^{-3}$, $12.5 \times 10^{-3}$, and $25 \times 10^{-3}$ wt%) are shown in Figure 4 (*n* = 3). Comparing Figure 4a–d, a maximum peak current difference (ΔI = 192.5 μA) is obtained at a concentration of $5.0 \times 10^{-3}$ wt%. The current differences decrease as the NiPc/THF concentration increases (Figure 4e), which is possibly due to the limited movement of electrons at higher NiPc/THF concentration coated on the gold electrode. Therefore, $5.0 \times 10^{-3}$ wt% of NiPc/THF was used as the optimal concentration for the preparation of the Au/NiPc electrode.

3.2.2. Optimal Conditions of BSA

In order to avoid non-specific bonding between the p53 antigen and NiPc film and to ensure that p53 antigens only bind to anti-p53, BSA was used as a blocking buffer. BSA concentrations of 1.36, 13.6, 136, 272, and 1360 μg/mL for blocking the Au/NiPc/anti-p53 (100 ng/mL) electrode were analyzed by DPV under 10 mM $K_3Fe(CN)_6$/($K_4Fe(CN)_6$ solution. The results are shown in Figure 5, where Figure 5f is a map of peak current changes (ΔI) in the oxidizing front of BSA at different concentrations (*n* = 3). The findings indicated that the amount of ΔI initially increased with BSA concentration and then became saturated for a BSA concentration of 272 μg/mL. In order to avoid the saturation of the BSA concentration to affect the measurement results, a BSA concentration of 136 μg/mL was selected as the optimal value for the blocking buffer.

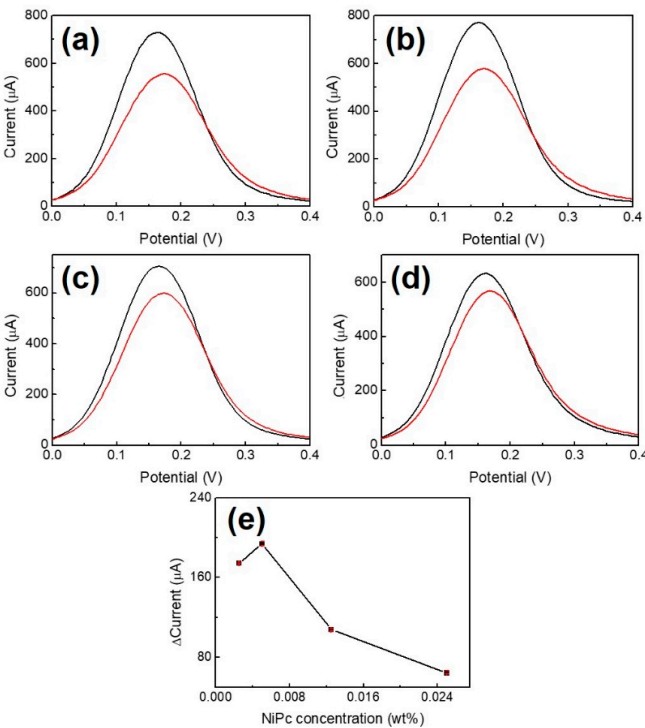

**Figure 4.** Differential pulse voltammetry (DPV) analysis for the preparation of Au/NiPc electrodes at different NiPc/THF concentrations. (**a**) $2.5 \times 10^{-3}$ wt%, (**b**) $5.0 \times 10^{-3}$ wt%, (**c**) $12.5 \times 10^{-3}$ wt%, and (**d**) $25 \times 10^{-3}$ wt%. The black and red curves represent the results of the Au and Au/NiPc electrode measurements, respectively. (**e**) Peak current change versus concentration of NiPc/THF ($n = 3$).

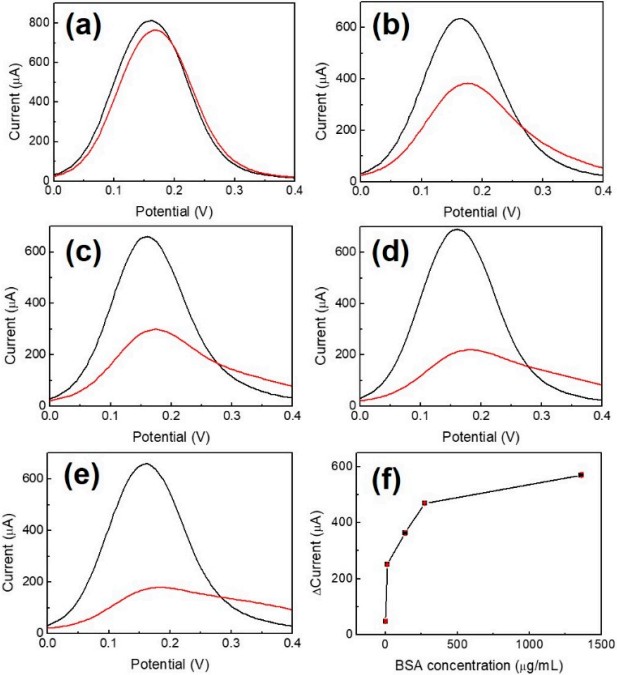

**Figure 5.** DPV measurement with BSA of different concentrations used as a blocking buffer (**a**) 1.36 μg/mL, (**b**) 13.6 μg/mL, (**c**) 136 μg/mL, (**d**) 272 μg/mL, (**e**) 1360 μg/mL, and (**f**) current change versus concentration of BSA (n-3). The black and red curves illustrate the results of Au/NiPc and Au/NiPc/BSA electrode measurements, respectively.

### 3.3. Electrochemical Sensing

Figure 6 shows the capacity of Au/NiPc/anti-p53 (1 ng/mL)/BSA and Au/NiPc/anti-p53 (100 ng/mL)/BSA electrodes to detect different p53 antigen concentrations. Original DPV data are provided in Figures S1 and S2 of the Supplementary Material. In Figure 6, the results of electrochemical sensing depict the current difference (termed Δcurrent) between the Au/NiPc/anti-p53/BSA electrodes bonded to p53 at different concentrations and the Au/NiPc/anti-p53/BSA electrode when the p53 concentration was zero (blank control). When the Δcurrent versus the p53 concentration was in a straight line, the Au/NiPc/anti-p53/BSA electrode was sensitive to p53 detection as the signal was directly related to p53 levels. The greater the slope of the straight line, the better the sensitivity. As for the Au/NiPc/anti-p53 (1 ng/mL)/BSA electrode (blue line), the linear range of p53 antigen detection was between 0.1 and 100 pg/mL. The linear relationship between the current (I) and the p53 antigen concentration ($C_{p53}$) was expressed by the following formula: I = 9.17 Log $C_{p53}$ + 9.62 ($R^2$ = 0.991). The sensitivity—calculated from the linear current slope divided by the area of the working electrode—was 38.60 µA/Log (pg/mL)-cm$^2$. With regard to the Au/NiPc/anti-p53 (100 ng/mL)/BSA electrode (red line), the linear range of p53 antigen detection was between 0.1 and 500 pg/mL. The linear relationship was as follows: I = 14.40 Log $C_{p53}$ + 18.05 ($R^2$ = 0.998). The calculated sensitivity was 60.65 µA/Log (pg/mL)-cm$^2$. These results indicate that the Au/NiPc/anti-p53 (100 ng/mL)/BSA electrode is characterized by a wide detection range and a high sensitivity in presence of elevated anti-p53 concentrations. A potential explanation for these findings is that a high concentration of anti-p53 on the electrode (100 ng/mL) can lead to a more efficient binding with the p53 antigen. Upon binding, the resulting current difference is large—ultimately resulting in a high sensitivity (60.65 µA/Log (pg/mL)-cm$^2$). In addition, a higher concentration of anti-p53 on the electrode engages a higher quantity of p53 antigens—this in turn would lead to a wider detection range (0.1–500 pg/mL). In reproducibility experiments (*n* = 3), different p53 concentrations (0.1, 10, 100, 500 pg/mL) were tested with the Au/anti-p53 (100 ng/mL)/BSA electrode. The corresponding relative standard deviations (RSDs) were 21.33%, 6.43%, 7.30%, and 2.76%, respectively.

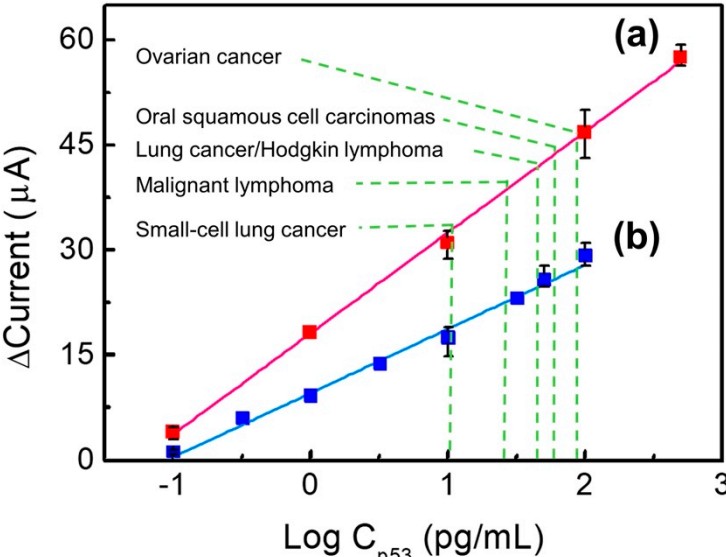

**Figure 6.** A comparison of the concentration of p53 antigens detected with Au/NiPc/anti-p53/BSA electrodes. The anti-p53 concentrations of the red line is 100 ng/mL (**a**) and for the blue line is 1 ng/mL (**b**).

Similarly, the RSDs for the detection of different p53 concentrations (0.1, 10, 50, and 100 pg/mL) with the Au/anti-p53 (1ng/mL)/BSA electrode were 30.36%, 13.88%, 6.41%, and 5.69%, respectively (*n* = 3). Taken together, these findings indicate that the Au/anti-

p53 (100 ng/mL)/BSA electrode ensured a higher reproducibility than the Au/anti-p53 (1 ng/mL)/BSA electrode. A comparison of different biosensors for the detection of p53 proteins is shown in Table 2.

**Table 2.** Comparison of different methods for detecting p53 proteins [14,16,29–35].

| Material | Method | Detection Range (µg/mL) | Preparation Time | Detection Time | Ref. |
|---|---|---|---|---|---|
| Silicon-on-insulator | Piezoresistive Readout | 0.02–20 | - | - | [29] |
| Silver | LSPR [a] | $> 5.9 \times 10^{-5}$ | >1 day | - | [30] |
| Graphene | Electrochemistry | $2 \times 10^{-4}$–$1 \times 10^{-3}$ | ~5 h | - | [31] |
| Graphene-gold | Electrochemistry | $1 \times 10^{-7}$–$1 \times 10^{-4}$ | ~7 h | - | [32] |
| Bi-SPCE [b] | Electrochemistry | $2 \times 10^{-5}$–$2 \times 10^{-2}$ | ~3 h | - | [33] |
| GCE/PEDOT:PSS/AuNPs/Ab$_1$ [c] (ZIF-8/DAP/Ab$_2$ [d]) | Electrochemistry | $1 \times 10^{-3}$–$1.2 \times 10^{-1}$ | ~12 h | 5 min | [14] |
| SPCE/PEI/NPs-Ab-BSA [e] | Electrochemistry | $1 \times 10^{-6}$–$1 \times 10^{-2}$ | ~2 h | - | [34] |
| Au/PEG/EDC/NHS-Ab/BSA [f] | Electrochemistry | 100–$1 \times 10^4$ pM | ~12 h | - | [35] |
| SPCE | Electrochemistry | $5 \times 10^{-3}$–$1.5 \times 10^{-1}$ | ~3 h | - | [16] |
| Au/NiPc/anti-p53/BSA [g] | Electrochemistry | $1 \times 10^{-7}$–$1 \times 10^{-4}$ | ~5 h | 90–150 s | This work |
| Au/NiPc/anti-p53/BSA [h] | Electrochemistry | $1 \times 10^{-7}$–$5 \times 10^{-4}$ | ~5 h | 90–150 s | This work |

[a] LSPR: Localized Surface Plasmon Reasonance; [b] Bi-SPCE: Bismuth coated Screen-Printed Carbon Electrode; [c] GCE/PEDOT:PSS/ AuNPs/Ab$_1$: Glassy Carbon Electrodes/Poly(3,4-Ethylenedioxythiophene): Polystyrene-sulfonate/Gold nanoparticles/Primary antibodies; [d] ZIF-8/DAP/Ab$_2$: Zeolitic Imidazolate Framework/2, 3-Diaminophenazine/Secondary antibodies; [e] SPEC/PEI/NPs-Ab-BSA: Screen-Printed Carbon Electrode/Polyethyleneimine/Carboxylated NiFe$_2$O$_4$ nano- particles-anti-p53-BSA; [f] Au/PEG/EDC/NHS-Ab/BSA: Gold/Polyethylene Glycol/N-ethyl-N-(3-(dimethylamino)propyl) carbodiimide/N-hydroxysuccinimide-anti-p53-BSA; [g] The amount of anti-p53 used to prepare the Au/NiPc/anti-p53/BSA electrode was 1 ng/mL; [h] The amount of anti-p53 used to prepare the Au/NiPc/anti-p53/BSA electrode was 100 ng/mL.

The results indicated that the detection system described in the current study has a lower detection limit and a larger detection range. Additionally, both the preparation process and the protein detection time were faster than the previously reported biochemical method [14]. The clinically significant cut-off levels for the detection of the p53 protein in the published literature are 90 pg/mL for ovarian cancer [36], 60 pg/mL for oral squamous cell carcinomas [37], 50 pg/mL for lung cancer [38] and Hodgkin lymphoma [35], 30 pg/mL for malignant lymphoma [38], and 10 pg/mL for small-cell lung [38]. Consequently, both the Au/NiPc/anti-p53 (1 ng/mL)/BSA and Au/NiPc/anti-p53 (100 ng/mL)/BSA electrodes successfully met the detection needs of p53 in these solid malignancies.

### 3.4. Specificity Test

The following compounds were examined in relation to their potential confounding effects on the sensor electrochemical behavior: 0.6 mg/dL AA, 5 mg/dL UA, 100 mg/dL glucose, 0.136 mg/mL BSA, 1.9 mg/dL histidine, 4 ng/mL PSA, 6.57 mg/dL HSA, and 1.0 mg/dL human IgG. A significance test analysis of the experimental data is provided in Figure 7 (*n* = 3). From Figure 7, ∆I of 500 pg/mL p53 antigen detected by the Au/NiPc/anti-p53/BSA electrode was 60 µA. Compared to other interfering compounds, the interference ∆I was extremely limited and the *p* value was less than 0.0005. These results indicate that the sensor electrochemical behavior is markedly different for the p53 antigen with respect of all other interfering compounds. Consequently, the Au/NiPc/anti-p53/BSA electrode described in this study can be considered substantially selective for the detection of p53.

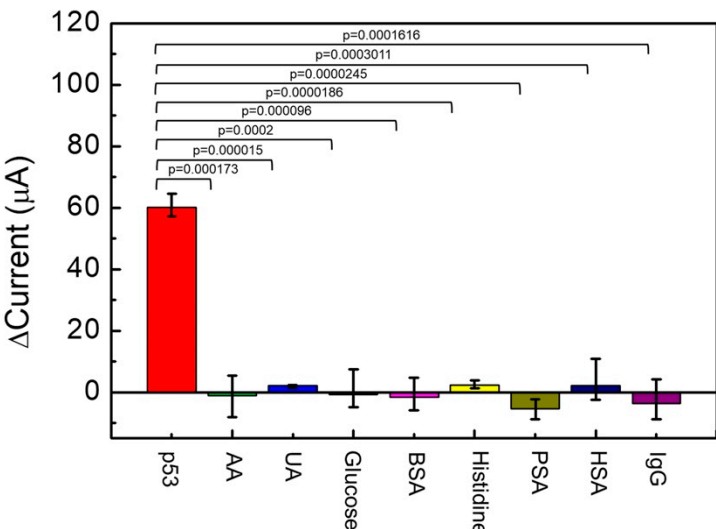

**Figure 7.** The specificity test on the detection of p53 antigens using the Au/NiPc/anti-p53/BSA electrode with the anti-p53 concentration of 100 ng/mL (*n* = 3).

## 4. Conclusions

The present study describes the electrochemical detection of the p53 antigen using Au/NiPc/anti-p53 (1 ng/mL)/BSA and Au/NiPc/anti-p53 (100 ng/mL)/BSA electrodes. The high affinity between NiPc and the gold electrode along with the ability of NiPc to bind proteins in a self-assembled manner allowed capturing the p53 antibody on the Au/NiPc surface. BSA was used as a blocking buffer to avoid non-specific binding between the p53 antigen and NiPc film; additionally, this approach ensured the specific formation of a complex between p53 antigens and the anti-p53 antibodies. The detection of the p53 antigen using the Au/NiPc/anti-p53 (1 ng/mL)/BSA and Au/NiPc/anti-p53 (100 ng/mL)/BSA electrodes under 200 μL of 10 mM $K_3Fe(CN)_6$)/($K_4Fe(CN)_6$ solution was analyzed by CV and DPV methods. The calibration line for p53 was constructed by analyzing the electrochemical current changes at different BSA and p53 antigen concentrations. The p53 antigen was detected with Au/NiPc/anti-p53 (100 ng/mL)/BSA electrodes through different optimization steps, with a detection range of 0.1–500 pg/mL, a linearity of $I = 14.403 \text{ Log } C_{p53} + 18.05$ ($R^2 = 0.998$), and a sensitivity of 60.65 μA/Log (pg/mL)-cm². Notably, the detection times of the p53 antigen using the study electrodes were remarkably fast (90–150 s). The Au/NiPc/anti-p53 (100 ng/mL)/BSA electrode offers numerous advantages, including high sensitivity, good specificity, and selectivity. The results of this study hold great promise and can pave the way to the development of a simple and highly accurate screening tool for clinical use in the real-time detection of p53 antigens. This sensor warrants further scrutiny for early cancer diagnosis and prognostication.

**Supplementary Materials:** The following are available online at https://www.mdpi.com/2227-9040/9/1/17/s1, Figure S1: DPV analysis of different p53 antigen concentrations as detected by Au/NiPc/anti-p53/BSA electrodes. The black, red, blue and green curves represent $5 \times 10^{-3}$ wt% NiPc, 1 ng/mL anti-p53, 136 μg/mL BSA, and (a) 0.1, (b) 0.32, (c) 1, (d) 3.2, (e) 10, (f) 32, (g) 50, and (h) 100 pg/mL of p53 antigen, respectively, Figure S2: DPV analysis of different p53 antigen concentrations as detected by Au/NiPc/anti-p53/BSA electrodes. The black, red, blue and green curves represent $5 \times 10^{-3}$ wt% NiPc, 100 ng/mL anti-p53, 136 μg/mL BSA and (a) 0.1, (b) 1, (c) 10, (d) 100, and (e) 500 pg/mL of p53 antigen, respectively.

**Author Contributions:** Conceptualization, M.-Y.H.; formal analysis, Y.-J.C. and Y.-R.P.; investigation, Y.-J.C., Y.-R.P., and H.-Y.L.; resources, M.-Y.H.; data curation, Y.-J.C.; writing—original draft preparation, Y.-J.C. and Y.-R.P.; writing—review and editing, M.-Y.H.; visualization, T.-Y.H.; supervision, C.-S.L. and M.-Y.H.; project administration, M.-Y.H.; funding acquisition, C.-S.L. and M.-Y.H. All authors have read and agreed to the published version of the manuscript.

**Funding:** This research was funded by the Ministry of Science and Technology of the Republic of China (MOST 109-2221-E-182-012 and MOST 107-2221-E-182-019), the Chang Gung Memorial Hospital, and the Chang Gung University (BMRP 576).

**Institutional Review Board Statement:** Not applicable.

**Informed Consent Statement:** Not applicable.

**Data Availability Statement:** Data sharing not applicable.

**Conflicts of Interest:** The authors declare no conflict of interest.

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
