# Peer review of "Preparation and Characterization of Au/NiPc/Anti-p53/BSA Electrode for Application as a p53 Antigen Sensor"

_chemosensors, doi:10.3390/chemosensors9010017_

Round 1

Reviewer 1 Report

Dear Editor,

This paper described a label free electrochemical immunosensor based on a modified gold electrode with nickel phthalocyanine (Au/NiPc) for the detection of tumor suppressor protein p53. Authors presented results about the performance of this sensor. Although these data are interesting, I do not consider that experimental results shown are enough for publishing this paper. Experimental data regarding important analytical characteristics, stability, selectivity and applicability of this new sensing platform should be presented.

Moreover, although the paper is well structured. Therefore, I do recommend considering the publication of this paper after addressing the following major concerns.

  1. 1. Apparatus and reagents: The Authors should indicate the area of the working electrode.
  2. 1. Apparatus and reagents: The Authors should indicate that the electrodes was used in the work were commercial screen-printed gold electrodes.
  3. 2. Film fabrication process: The Authors should indicate what methodology was used for the synthesis of NiPc
  4. 2.2 Preparation of Au/NiPc electrode: The Authors should indicate if the working gold electrode were previously washed and what procedure was used to perform the preparation of the electrode surface.
  5. 4. Detection of p53 concentration: Why do the Authors do the incubations at 4ºC? why not 25 or 37ºC where the kinetic of the immunoreactions is faster?
  6. 5. Specificity test of p53 by Au/NiPc/anti-p53/BSA electrode: In this kind of studies the Authors should include hemoglobin, HSA and human IgG, which are the majority compounds in human serum.
  7. Figure 4, 5 and 6, the Authors should include error bars.
  8. 3 Electrochemical sensing: In the calibration plots the Authors should change I for Δi.
  9. 3 Electrochemical sensing: The Authors should include studies of reproducibility of the develop immunosensor (RSD, %).
  10. 3 Electrochemical sensing: “…detection time are faster than the other bio-chemical methods [29-33] …” In the Table there is not the times for others works so The Authors cannot say this affirmation.
  11. 3 Electrochemical sensing: Why do not the Authors use more concentration of the capture Ab?
  12. Table 2: The Authors should include more recently references for the detection of p53 with electrochemical immunosensors: (DOI: 10.1016/j.bioelechem.2020.107647; 10.1007/s00604-020-04594-z; 10.3390/s20216364; 10.1016/j.aca.2018.08.023; 10.3390/bios6040056; 10.1016/j.trac.2017.01.007…)
  13. The Authors should include studies of the practical applicability of the develop immunosensor (sample analysis).

Other minor mistakes to be corrected:

  • 1.1 Ultraviolet-visible and near-infrared (UV-Vis near IR) analysis: exchanging, format
  • Conclusions: “… The calibration line of the p53 antigen was created by analyzing the electrochemical current changes for various concentrations of cAb and p53 antigen …”

 Sincerely,

                  The reviewer

Reviewer 2 Report

The article is detailed enough and the conclusions are supported by the presented data. Although the exact subject is novel (a new type of electrode for the p53 protein concentration), the broader subject (the p53 protein and detection by electrodes) is not new.

The graphical elements are good and clearly enough to read and understand the presented data.

Reviewer 3 Report

In this paper, a nickel phthalocyanine modified gold electrode was fabricated and used for the immunodetection of p53 protein. The goal of the article is interesting, but the work is not sufficiently elaborated and there is still some work to be done on it in order to demonstrate that the proposed method is a good alternative for quantifying p53 protein. 1. What are the red and black curves representing for in Figure 5? 2. How did the authors perform the relevant experiment and calculate â–³I in Figure 6? Original DPV data with different concentrations of p53 antigens should be provided. 3. Error bars in Figure 6 are missing. 4. What are the repeatability, reproducibility and stability of the sensor? 5. There are a few typos scattered throughout the text.

Round 2

Reviewer 1 Report

Dear Editor,

This paper described a label free electrochemical immunosensor based on a modified gold electrode with nickel phthalocyanine (Au/NiPc) for the detection of tumor suppressor protein p53. The Authors have answered all major concerns in the present version of the work, so I do recommend considering the publication of this paper in the present version.

Sincerely,

                  The reviewer

Reviewer 3 Report

The authors have replied most of the questions. The manuscript can be published in its present form.